# Self-Powered Synchronized Switching Interface Circuit for Piezoelectric Footstep Energy Harvesting

**DOI:** 10.3390/s23041830

**Published:** 2023-02-06

**Authors:** Meriam Ben Ammar, Salwa Sahnoun, Ahmed Fakhfakh, Christian Viehweger, Olfa Kanoun

**Affiliations:** 1Measurements and Sensor Technology, Faculty of Electrical Engineering and Information Technology, Chemnitz University of Technology, 09126 Chemnitz, Germany; 2National School of Electronics and Telecommunications of Sfax, University of Sfax, Sfax 3038, Tunisia; 3Laboratory of Signals, Systems, Artificial Intelligence and Networks, Digital Research Center of Sfax, University of Sfax, Sfax 3038, Tunisia

**Keywords:** wearable energy harvester, wearables, energy management, piezoelectric energy harvesters, self-powered, synchronized switching, vibration, human motion, footsteps, rectifiers, piezoelectric insole, piezoceramics, power supply

## Abstract

Piezoelectric Vibration converters are nowadays gaining importance for supplying low-powered sensor nodes and wearable electronic devices. Energy management interfaces are thereby needed to ensure voltage compatibility between the harvester element and the electric load. To improve power extraction ability, resonant interfaces such as Parallel Synchronized Switch Harvesting on Inductor (P-SSHI) have been proposed. The main challenges for designing this type of energy management circuits are to realise self-powered solutions and increase the energy efficiency and adaptability of the interface for low-power operation modes corresponding to low frequencies and irregular vibration mechanical energy sources. In this work, a novel Self-Powered (SP P-SSHI) energy management circuit is proposed which is able to harvest energy from piezoelectric converters at low frequencies and irregular chock like footstep input excitations. It has a good power extraction ability and is adaptable for different storage capacitors and loads. As a proof of concept, a piezoelectric shoe insole with six integrated parallel piezoelectric sensors (PEts) was designed and implemented to validate the performance of the energy management interface circuit. Under a vibration excitation of 1 Hz corresponding to a (moderate walking speed), the maximum reached efficiency and power of the proposed interface is 83.02% and 3.6 mW respectively for the designed insole, a 10 kΩ resistive load and a 10 μF storage capacitor. The enhanced SP-PSSHI circuit was validated to charge a 10 μF capacitor to 6 V in 3.94 s and a 1 mF capacitor to 3.2 V in 27.64 s. The proposed energy management interface has a cold start-up ability and was also validated to charge a (65 mAh, 3.1 V) maganese dioxide coin cell Lithium battery (ML 2032), demonstrating the ability of the proposed wearable piezoelectric energy harvesting system to provide an autonomous power supply for wearable wireless sensors.

## 1. Introduction

The design of wearable electronics has evolved in recent years towards miniaturization, integration, and connectivity thanks to advancements in the fields of micro-machining, self-powered integrated circuits, and micro-electro-mechanical systems (MEMS) [1,2,3]. This resulted in an increasing need for autonomous power supplies [4,5]. On the other hand, the power consumption of wearable sensors has steadily dropped to reach even the micro-watt range [6,7]. Rechargeable batteries are the primary and most common energy source for most of these devices, and if they have a lightweight, they have consequently lower battery capacities. Batteries limit the time of operation and lead to discontinuous sensing and transmission of the data [8]. Hence, they require frequent battery recharging or changing, which is undesirable. In particular, in medical wearable applications, replacing or charging a battery may require professional assistance depending on the state of the patient. All these aspects have driven the design of wearable energy harvesters, which harvest the energy related to human body state or movements to produce a sustainable energy supply and provide an alternative to the inconvenient use of batteries [9]. Wearable energy harvesters can be based on several mechanisms, including electromagnetic, electrostatic, thermoelectric, nano-triboelectric, and piezoelectric [10,11,12,13] principles. Piezoelectric converters have numerous benefits over electromagnetic and electrostatic ones, such as simple construction, high output power density, and great integration with micro-electro-mechanical systems (MEMS) [14,15,16]. Piezoelectric materials produce an electric charge under applied forces. However, some factors limit the efficiency of piezoelectric materials in energy harvesting applications. The produced charge is quickly neutralized by the inner resistances and parasitic effects of piezoelectric materials. Additionally, piezoelectric converters produce high voltage levels and are highly susceptible to noise. Therefore, sophisticated energy management circuits are required to efficiently extract produced charges and to filter out related noise to use the harvested energy for power supply applications [17]. PEts can be electrically modelled as sinusoidal current sources ($I_{p}$) in parallel with their internal parasitic capacitances ($C_{p}$) as shown in Figure 1.

The total amount of charge ($Q_{P}$) produced during a single deformation event is proportional to the deformation applied to the PEt. The applied deformation force to the PEt results in the generation of the current $I_{p}$ and charges the internal capacitance $C_{p}$. An electronic interface is necessary to ensure voltage compatibility between the piezoelectric element and the load because the piezoelectric transducer can deliver irregular alternating current (AC) rather than direct current (DC) [5]. The effectiveness of energy harvesting is significantly influenced by the electronic interface, which has sparked numerous research initiatives to design suitable and effective Piezoelectric energy harvesting (PEH) interfaces [18,19,20,21,22]. The main goal of implementing these circuits is to extract maximum energy from piezoelectric harvesters, efficiently manage the energy within the system, and thus enable the user to supply DC loads such as wearable sensors. However, in recent years, there has been a trend toward energy harvesting from high open-circuit voltage ($V_{O}C$) excitations produced by PEts [23,24,25]. The contributions of this paper are:We provide an up-to-date literature review of the state-of-the-art electrical interfaces for piezoelectric energy harvesting, including a detailed analysis of the architecture and working principle of each circuit.We propose a novel self-powered energy harvesting interface (SP-PSSHI) able to harvest energy from irregular human footsteps and efficiently manage the extracted power to be stored for future use.We designed a wearable piezoelectric energy harvesting system composed of 3 main blocks: the designed shoe insole with integrated PEts to serve as a harvester for the system; the proposed energy harvesting interface; and a coin cell battery to perform energy storage and battery charging actions.We demonstrate the performance of the system and we validate it with a detailed experimental study.

The paper is structured as follows: Section 2 surveys the state-of-the-art electrical interfaces for piezoelectric energy harvesting interfaces. Section 3 and Section 4 present the proposed circuit, detailed explanations, related analytical investigations and simulation results. Section 5 concerns the experimental validation of the proposed circuit. Finally, Section 6 gives a summary and concludes the paper.

## 2. State of the Art of Synchronised Switching Based Electrical Interfaces for Piezoelectric Energy Harvesting

One main limitation of conventional AC-DC energy harvesting circuits for PEHs is that negative output power is generated since the output current and voltage could not maintain the same phase [17,26,27]. This leads to a significant power loss of the harvested energy. A full wave bridge rectifier, also known as the standard energy harvesting (SEH) approach, was the first step to implementing a self-powered energy harvester. It is the most widely used technique for PEHs with a simple uncontrollable architecture [28]. When a positive potential is generated from the piezoelectric harvester, the full bridge rectifier will rectify the voltage into DC voltage. On the other hand, when a negative potential is generated from the PEH, the voltage will also be rectified into a DC output. The disadvantages of such a standard architecture are load dependency, low efficiency, and high losses due to the phase difference between the current and voltage. Synchronized Switch Harvesting on Inductor (SSHI) technology was adapted to overcome these limitations by applying a resonance with the association of an inductor and a switch with the PEH, establishing an oscillating LC circuit with the internal capacitor of the harvester. This resonance effect influences the amount of charge present in the parasitic piezoelectric capacitance, resulting in a maximum voltage being generated and thus increasing the extracted power. When dealing with this technology, two main interfaces can be reviewed at the first stage, Parallel Synchronized Switch Harvesting on Inductor (P-SSHI) or Serial synchronized Switch Harvesting on Inductor (S-SSHI). The switch S and inductor L are either connected in series to address S-SSHI or in parallel to address P-SSHI. Figure 2a,b represents the interface circuit of P-SSHI and S-SSHI respectively.

The working principle of the P-SSHI solution, as studied by Nechibvute in [21], relies on inverting the energy after the extraction process by controlling the switch S1, and the inductor L1 [30]. When the vibration occurs, the switch S1 remains open, allowing the current to flow through the circuit to the storage element $C_{str}$. If the piezoelectric transducer’s voltage drops below a certain threshold, the switch S1 will automatically close, inverting the PEH voltage and therefore stopping the current flow. This means that the switch is kept closed until a full inversion of the PEH’s voltage has been achieved. Furthermore, the inductor’s integration reduces the phase difference between the voltage and current, thus increasing the harvesting efficiency. Nevertheless, this voltage inversion causes electrical damping that opposes the mechanical vibrations on the piezoelectric material. This effect is known as Synchronized Switch Damping (SSD) [31]. It can significantly affect the overall conversion efficiency, and it is consequently the main limitation of both P-SSHI and S-SSHI circuits [19].

Synchronized Electrical Charge Extraction (SECE) interface is another PEH solution proposed by Guyomar et al. in [18], which involves an inductance as an energy storage component. Thus, the harvesting process is carried out in two steps. First, the energy available on the PEH is transferred to the inductor. Then the PEH is disconnected from the circuit and the energy stored in the inductor is transferred to the storage capacitor. This solution prevents mainly the SSD effect, which is the main limitation of P-SSHI and S-SSHI circuits [32]. This effect is caused by the direct connection between the output load and the PEH during the whole vibration phase. When the PEH generates the voltage, the switch S will be closed, and the energy will be stored in the inductor L as seen in Figure 2c,d.

When the vibration stops, the piezoelectric element voltage falls to zero, and the switch S will open immediately. Consequently, the energy accumulated in the inductor will be directly transferred to the storage capacitor and the load. One limitation of this interface is the complexity when compared to the simple architecture and switching strategy that characterize SSHI circuits [29].

Lallart et al. [33], also proposed another hybrid energy harvesting scheme by combining the SSHI architecture with the SECE interface. It is named Double synchronized Switch Harvesting on Inductor (D-SSHI). In this approach, the circuit behaves as follows; first transferring a part of the harnessed energy from the PEH to an intermediate storage capacitor, $C_{int}$, and using the remaining energy for the inversion process. Then, transferring the energy in $C_{int}$ to the inductance and finally to the load as depicted in Figure 2e,f.

The D-SSHI solution is less sensitive to load changes among the reviewed solutions thanks to the indirect connection between the PEH and the load. But due to extra components used, it has the highest energy consumption. The complexity level of this circuit is also considered very high due to the double-switching mechanism which let the solution more suitable for high-power applications [18,31].

The control of the integrated switches was a common limitation for the reviewed interfaces, so several researchers were focusing on developing self-powered resonant energy-harvesting circuits. The frequently used switches are realized either by Bi-polar Junction Transistor (BJT) or Metal Oxide Semiconductor Field Effect Transistor (MOSFET). Therefore, the performance difference comes mainly from three aspects: the driven mechanism, the turning-on threshold and the gate parasitic capacitance. The reviewed interface [30] is based on a Peak Detector (PKD) circuit used to detect the maximum and minimum input signal of the PEt and drive the electronic switches synchronously without providing any external power. The topology relies on electronic breakers named also as complementary transistors topology. Similar to the SSHI working principle, switch S1 will be controlled by sensing the maximum and minimum voltage of the PEH. It is named an envelope voltage detector composed of the path through R2, D5 and C1 as seen in Figure 3. It detects the maximum and minimum voltage generated from the PEH. When the voltage generated decreases from its maximum threshold value, the switch S1 will be closed and immediately inverts the voltage via the inductor. Thus, the path through D7, Q2, and L1 is referred to as the “digital switch path” (see Figure 3). The main limitation of this interface is the energy consumption caused by the forward voltage of the integrated diodes and the current-driven switching process caused by the use of Bi-polar transistors as switches.

To overcome this limitation, a recent scheme, referred to as NVC-PSSHI [34], was introduced for PEH. The interface is mainly composed of a Parallel Synchronized Switch Harvesting on Inductor (P-SSHI) and a Negative Voltage Converter (NVC). In this interface, the switching is a voltage driven-mechanism realized by two MOSFETs. The targeted input parameters included 3 Volts peak to peak (Vpp) to 7 Vpp input voltage, 100 Hz to 500 Hz frequency range, and 5 kΩ to 30 kΩ DC loading conditions. The proposed interface was up to 23.4% more efficient than the standard P-SSHI interface. Nevertheless, the current flow to the inductive path in the P-SSHI is limiting the energy flow from the PE harvester to the load and thus decreasing the efficiency of the storage process. In addition, the proposed interface was not evaluated against irregular input excitations, especially in the case of energy harvesting from high open-circuit voltages ($V_{OC}$) excitations produced by PEts.

In SSHI-based PEH systems, a study of an adequate rectification marked in green in Figure 3 is extremely useful to enhance the interface efficiency. The diode bridge rectifier is shown in Figure 4a. This is the most used rectification circuit in SSHI interfaces for high and medium-power applications. A big disadvantage of this circuit is its low efficiency, which is due to the voltage presented by the diodes when a forward polarization is obtained. This can influence the performance of the whole SSHI circuit [35]. To boost the efficiency in ultra-low and low-power applications, the junction diodes of a traditional bridge rectifier are replaced by MOSFETs in a diode-tied topology (DMOS) adopting a conventional CMOS fabrication process [27]. The MOSFET bridge rectifier is an alternative to a diode bridge with diodes replaced by DMOS-connected MOSFETs consisting of 2 P channel MOSFETs (PMOS) (MP1 and MP2) and two N channel MOSFETs (NMOS) (MN1 and MN2) as seen in Figure 4b. The DMOS connection is realized by short-circuiting the gate and the drain terminals of the transistor, which maintains the transistor in the saturation region of operation [28]. A modular circuit of the MOSFET bridge is as presented in Figure 4b. Although this circuit presents significant improvements, especially in the frequency response of the rectifier, the rectifier still presents the efficiency-related disadvantages shown by the diode bridge rectifier due to the developed voltage between the drain and source terminals of each device (higher than the threshold voltage $V_{th}$ [36].

To enhance the Power Conversion Efficiency (PCE) in an integrated MOSFET-based full bridge operation, gate cross-coupled rectifier technology is considered the most popular solution [37]. This topology is widely used in high-frequency inductive coupling devices (see Figure 4c). The circuit is named Gate Cross-Coupled Rectifier (GCCR). It presents a great reduction of the voltage drop presented by the rectifier due to the connection of the upper transistor gates directly to the input voltage [38]. This sets a higher voltage at the gate terminals of the transistors [27], reducing the voltage between the drain and source $V_{DS}$ of the device. The circuit consists of 2 PMOS (MP1 and MP2) and 2 NMOS (MN1 and MN2) configurations where MN1 and MN2 are cross-coupled and MP1 and MP2 are in DMOS connections. Considering the positive interval of the AC voltage $V_{AC}$, the circuit operation is as follows. When the input voltage is lower than the threshold voltage of MN1 ($V_{AC}$ < $V_{th}$), there is no current flow, and the input and output ports are isolated from each other. When the input voltage exceeds the threshold voltage of MN1 ($V_{AC}$ > $V_{th}$), MN1 is turned on, connecting to the output node. When the input voltage exceeds the output voltage by more than the threshold voltage of MP2 ($V_{AC}$ > $V_{out}$ + $V_{th}$), the current flow is established by the commutation of the DMOS transistor MP2. A Cross coupled MOSFET-based circuit can also provide a virtual negative resistance in the circuit. Due to the direct connection of the upper transistor gates to the input voltage, the voltage drop presented by the rectifier is greatly reduced in GCC topology. This circuit’s efficiency has been estimated to be between 70 and 75%.

Other authors proposed the use of an alternative to the GCCR with SSHI interfaces, named negative voltage converter (NVC) [27,34,39]. It enables the rectification process to be carried out with very small voltage drops in the current path. The modular circuit is displayed as seen in Figure 4d. When the input voltage is greater than the transistor thresholds voltage ($V_{AC}$ > $V_{th}$), MN1 and MP2 conduct, allowing the current to flow. In the negative cycle of the input voltage, the working principle is similar to the one described previously, the only difference is that the transistors MN2 and MP1 provide the current path. The circuit’s commutation is only dependent on the input voltage rather than the link between the input and output voltages, as is the case with other circuits. Due to this property and the assumption that MOSFETs are bidirectional devices, currents flow from the output to the input whenever the output’s voltage is greater than the input voltage’s magnitude. If the input current is not controlled, this could result in a decreased rectifier efficiency [40].

Active rectification provides an additional way to further boost a rectifier’s efficiency by decreasing the conduction losses of the transistors acting as switches [37]. An active rectifier with cross-coupled PMOS switches is depicted schematically in Figure 4e. A comparator and related control circuitry drive the MOSFET in an active rectifier configuration. The MOSFET and its comparator are often referred to as an “active diode” [41]. Based on the characteristics of the source and the demands of the load, the comparator continually monitors the voltage drop of the MOSFET and turns it ON or OFF in order to minimize conduction loss and reverse leakage current. The selection of the rectifier topology has great importance and depends mainly on the system load [37].

## 3. Novel Approach of SP-PSSHI Piezoelectric Energy Harvesting Interface

The state-of-the-art P-SSHI solutions are always targeted toward extracting more current from PEHs. However, at a lower frequency, the impedance provided by the inductor ($L_{P-\mathrm{SSHI}}$) is low, leading to a high current passage through the inductor path, which limits the energy flowing into the storage element. Therefore, a novel Self-Powered P-SSHI for piezoelectric footstep energy harvesting interface is proposed to handle irregular footstep input excitations and to allow more energy flow to the storage device in order to drive DC loads. A simplified circuit diagram of the proposed piezoelectric harvesting interface is shown in Figure 5. It consists of 4 major blocks.

Piezoelectric Energy Harvester (PEH) and low pass filterPeak Voltage Detector and Triggering CircuitModified Resonant (P-SSHI) CircuitGCC-based AC/DC rectifier

In the next subsections, we will analyse the different blocks by describing the working principle as well as the functionality of the block in the proposed interface. The block “Modified Resonant (P-SSHI) circuit” will be detailed in a seperate Section 4.

### 3.1. Piezoelectric Energy Harvester and Low Pass Filter

There are different types of PEts. They can be either flexible or rigid. The flexible one generates an electric signal when stress is applied in the form of bending, it uses mainly polymers that are highly flexible, easy to process, and most importantly, biocompatible with the human body [2,42,43]. Their main disadvantage is that they have a low piezoelectric response, resulting in relatively low power extraction. The rigid type doesn’t necessarily bend, but with microscopic bending, it generates an electric signal, so it can be used to harvest energy. Piezoelectric diaphragms and cantilever structures are examples of rigid piezoelectric harvesters based on ceramic. Ceramic piezoelectric sensors produce a high alternating voltage in response to applied dynamic pressure or vibration. They can detect forces in the Z axes, generating electrical impulses with high amplitudes. They are simple to manufacture and, therefore, cheap. In recent years, piezoelectric circular diaphragm (PCD) energy harvesters have attracted considerable attention among the existing piezoelectric generators used for PEH [44,45]. In particular, in the case of footsteps energy harvesting, PCD energy harvesters are highly recommended [46] and thereby they were chosen for the carried experiments. When compressed, the harvester generates an electric signal that can be harvested, efficiently managed, and stored for later use. Referring to Thevenin’s theorem, the PCD can be electrically modelled as a voltage source in series with a capacitor as seen in Figure 5. The voltage generated by the harvester can be expressed in Equation (1):(1)$$V_{harvester}=V_{harvester}\sin\left(\omega t\right)$$

Pulses generated by a PCD attached to a human body are irregular. This reveals that the voltage generated can also be random and can exceed the maximum voltage tolerated by the CMOS process used for IC fabrication. Additionally, these harvesters produce high voltage levels and are highly susceptible to noise. Particularly, in the case of irregular footsteps input excitations, we proposed adding a low-pass filter to avoid any high-frequency input. The third-order low-pass filter is composed of two capacitors and an inductor. This design aim of the filter is to ensure that the high-frequency noise is reduced sufficiently to generate an accurate control signal with minimum phase lag and thus keep the efficiency of the wearable PEH system stable. Figure 5 shows the proposed system.

### 3.2. Peak Voltage Detector and Self Powered Power Supply Circuit

After filtering out the undesirable high-frequency noise signals derived from irregular footsteps, the high-frequency-attenuated inputs are in a second stage fed to a triggering circuit. The Peak Voltage Detection (PKD) of the PEH is realized via a differentiator circuit which senses the slope of the harvester capacitor’s voltage (*dV*/*dt*) and detects the maximum and minimum of the input voltage from the PCD to determine the triggering time based on $R_{trig}$ and $C_{trig}$ values. Figure 6 illustrates the PKD and triggering concepts with a highlight on the current flow within the proposed solution.

The SP power supply circuit consists of a comparator supply circuit (D3, D4, C4, C5), the differentiator circuit ($R_{trig}$ and $C_{trig}$) and a signal comparator unit ($C_{comp}$ and *comp*). The comparator supply circuit provides the positive and negative DC voltages sources, $V_{cc}$ and $-V_{cc}$ through the diodes (D3 and D4) and the capacitors (C4 and C5) to power the selected ultra-low-power comparator and the switches of the proposed circuit. The two diodes D3 and D4 maintain the correct positive and negative currents flows to charge the capacitors C4 and C5. C4 regulates the positive voltage between $V_{cc}$ and ground, and C5 regulates the negative voltage between $-V_{cc}$ and ground. The two regulated voltages, $V_{cc}$ and $-V_{cc}$ are then connected to the comparator, and the output of the comparator is used to drive the switches. Based on the components ($R_{trig}$ and $C_{trig}$), the comparator sends high and low output pulses to the P-SSHI unit. When the comparator output is low M2 (P channel) and D2 are conducting. When the comparator output is high M1 and D1 are conducting. The output voltage from the PKD circuit $V_{d}\left(t\right)$ is given by Equation (2):(2)$$V_{d}\left(t\right)=R_{trig}C_{trig}\frac{dV_{in-PSSHI}\left(t\right)}{dt}$$
where $R_{trig}$ is the triggering resistance, $C_{trig}$ is the triggering capacitance, and $V_{in-PSSHI}$ is the input voltage to the P-SSHI circuit. The comparator input voltage $V_{in-comp}$ is given by Equation (3), where $V_{p}$ is the peak input voltage and $V_{D}$ is the voltage drop related to the diodes D3 or D4.
(3)$$V_{in-comp}=\pm \left(V_{p}-V_{D}\right)$$

The comparator chosen here is a nano-power comparator from Texas Instrument, easy to drive and suitable for low power circuit design. The operation of the resonant circuit is based on the fact that the inductor and the PCD output capacitor can form a resonant network that is activated by closing switches (SW1) or (SW2). The switches are closed at the end of each half cycle and thus the voltage through the capacitor will immediately change polarity.

### 3.3. Gate Cross-Coupled Based AC-DC Rectifier

The below Table 1 [28] shows the performance of different simulated rectifier architectures with a pure resistive load of 3 kΩ. This study is carried out in order to evaluate the effect of the type of load on the rectifiers’ efficiencies. For RC loads, common in PEH, the GCCR should be selected which attains the highest efficiency with less sensitivity to load changes. It also maintains a relatively constant efficiency for both considered loads. Thus, GCCR is considered the most popular choice to increase the PCE in MOSFETs-based rectification circuits for SSHI solutions.

Another simulation detailed by Table 2 [28] shows the performance of different simulated rectifier architectures with low input voltage and an RC load at the output of the circuit of 3 kΩ and 3.3 mF respectively.

It can be observed from the above simulation tables that GCCR has better efficiency when compared to any other rectifier when the load is R-C in nature. However, when the load is purely resistive (R load), NVC provides the highest efficiency. It should be noted that the GCCR and MOSFET Bridge are connected via DMOS connection in the bridge circuit’s lower half. This is realized by short-circuiting the transistor’s gate and drain terminals, which maintain the transistor in the saturation region of operation $(V_{ds}>V_{gs}-V_{th}$). Therefore, each DMOS transistor is never fully on or fully off. This could increase the transistor voltage and decrease the efficiency of the whole SSHI interface.

To overcome the limitations related to the use of the NVC rectifier with P-SSHI circuits, in the case of RC loads, the adopted rectifier in our proposed circuit is based on GCC topology, with slight modifications on the lower half of the bridge. Such consideration is made mainly because GCCRs not only rectify but also help in providing virtual negative resistance in the circuit. Thus, the overall impedance of the circuit decreases which leads to a reduction in the circuit losses. This helps in increasing the output power and hence the overall efficiency of the PEH system increases. Figure 5 shows the MOSFETs connection in the rectifier unit. The MOSFETs M3 and M4’s gates are cross-coupled (P channel) and the lower half of MOSFETs M5 and M6 (N channel) are connected such that their respective sources are connected to the gates. The improvement over the GCCR is mainly done to avoid DMOS connection losses. The DMOS connection is realized by short-circuiting the gate and the drain terminals of the transistor, which maintain it in the saturation region of operation ($V_{ds}>V_{g}s-V_{T}$). Thus, each transistor is never fully on or fully off. The adopted connection allows the MOSFETs to be in on condition and virtually act as a diode with a low voltage drop which enhance the efficiency of the overall SP-PSSHI circuit.

In the positive cycle of the AC input, MOSFETs M3 (PMOS) and M6 (NMOS) are conducting allowing the positive half cycle of the voltage to be rectified and to be flowed to the load. In the negative half cycle of the AC, MOSFETs M5 (NMOS) and M4 (PMOS) will conduct (see Figure 5). This leads to a positive voltage being rectified through the load. Thus, leading to a pulsating DC output in both cycles. Hence, to have a smoother DC output (constant DC), a capacitor is used. The capacitance value is selected such that its discharging rate is slower so as to have a constant voltage over both positive and negative cycles of the input.

## 4. Modified Resonant (P-SSHI) Circuit

The P-SSHI circuit consists of an inductor connected in series with a resistor with M1 and M2 MOSFET switches (M1 and M2 are IRLML6426 and AO3435 respectively). The N channel switch is turned on when the triggering circuit output is high, while the P channel switch is turned on when the triggering circuit output is low. The current flowing through the Inductor (P-SSHI path) is very high when compared to the Load. Thus, the current through the inductor is approximately the same as the input inferring that the current through the load is very low which is confirmed by the simulation results shown in Figure 7a.

This is mainly because the impedance offered at the load side is very high when compared to the P-SSHI path resulting in the high current passage at the inductor. Therefore even if the inductance value is varied at a range of 100 mH to 1 H, the impedance is very low when compared to a 10 kΩ load resistance. To overcome this limitation, we propose adding a resistance connected in series with the inductor of the P-SSHI path to allow an impedance matching between the load and the input and provide a high resistance path at the P-SSHI unit. A comparison between Figure 7a,b shows the load current variation due to the inclusion of $R_{series}$. It can be seen that the load current without series resistance was found to be around 65–70 μA, but after the inclusion of the series resistance, the load current can be visualized at 550 to 600 μA (see Figure 7). The principal aim of such a proposal is to avoid the high current passage through the inductor of the P-SSHI unit by connecting a series resistance and consequently enhancing the energy flow to the storage unit and the power extraction ability at low frequencies to provide an autonomous power supply for DC loads.

According to the resonance principle, the resonant frequency is directly dependent on *L* and *C* through Equation (4).
(4)$$f_{resonance}=\frac{1}{2\pi \sqrt{LC}}$$

At resonance, the circuit reactance $X_{T}$ can be written as shown by Equation (5)
(5)$$X_{T}=X_{L}-X_{C}$$
(6)$$X_{L}=2\pi fL_{PSSHI}$$
(7)$$X_{C}=\frac{1}{2\pi fC_{Harvester}}$$

Therefore, the new equivalent circuit impedance of the P-SSHI circuit can be expressed by Equation (8) where $X_{L}$ is the inductive reactance of $L_{psshi}$, $X_{C}$ is the capacitive reactance of the harvester and f is the frequency of the applied vibration.
(8)$$Z=\sqrt{R_{series}^{2}+X_{T}^{2}}$$

The below Equation (9) can be used to determine the inductance value $L_{PSSHI}$.
(9)$$\pi \sqrt{L_{PSSHI}\;\cdot \;C}\ll \frac{T_{S}}{2}$$
where $T_{S}$ is the mechanical oscillation period. It is assumed that the mechanical displacement of the PCD $X_{m}\left(t\right)$ is sinusoidal, the related velocity is noted as $\dot{X(t)}$ and the equivalent current generated from the PCD is $I_{eq}$. They can be respectively expressed by Equations (10)–(12). α is the piezoelectric coefficient force related to the conversion of the mechanical force to an electrical signal.
(10)$$X_{m}\left(t\right)=-\hat{X_{m}}\cos\left(\omega t\right)$$
(11)$$\dot{X_{m}\left(t\right)}=\omega \hat{X_{m}}\sin\left(\omega t\right)=\hat{\dot{X_{m}}}\sin\left(\omega t\right)$$
(12)$$I_{eq}\left(t\right)=\alpha \dot{X_{m}}=\alpha \omega \hat{X_{m}}\sin\left(\omega t\right)=\hat{I_{eq}}\sin\left(\omega t\right)$$
where $\left\{ \begin{array}{cccc} & \hat{\dot{X_{m}}} & = & \omega X_{m}\\ \hat{I_{eq}} & & = & \alpha \omega \hat{X_{m}} \end{array}.\right.$

When the voltage from the *PCD*
$V_{PCD}$ drops to a certain value $-\hat{V_{c}}$, *X* equals $-\hat{X}$ and the current $I_{eq}$ drops to zero as well, that time is referred as $T_{1}$. The time when the *PCD* capacitor and the PSSHI inductor *L* resonate in a half resonant cycle and $V_{PCD}$ equals $\hat{V_{c}}$.γ is noted $T_{2}$. $T_{2}$ equals $\frac{1}{2}$$T_{LC}$ which is the *LC* resonant period. γ is defined as the inverting quality factor *Q* of the *LC* resonant circuit of the proposed interface as shown by Equations (13) and (14).

$R_{LC}$ in the expression of the quality factor *Q* is considered as an electrical loss in the system. The expressions of γ and *Q* are given respectively by the following Equations (13) and (14). The time when $V_{PCD}$ equals $\hat{V_{c}}$ is referred as $T_{3}$.
(13)$$\gamma =e^{-}\frac{\pi }{2Q}$$
(14)$$Q=\frac{1}{R_{LC}}\sqrt{\frac{L_{PSSHI}}{C_{harvester}}}$$

t∈ [$T_{1}$, $T_{2}$]The half *LC* resonant period $\frac{T_{LC}}{2}$ is equal to the time interval between $T_{1}$ and $T_{2}$. During this interval, the inductor of the PSSHI interface will resonate with the internal capacitor of the *PCD*. Consequently, the terminal voltage of the piezoelectric transducer reverses its polarity from the negative voltage to the positive voltage. Equation (15) can be used to express the terminal voltage of the *PCD* during the oscillating period. The terminal voltage at time $T_{2}$ can be calculated as shown by Equation (16) by substituting $t=\frac{T_{LC}}{2}$ in Equation (15).



(15)
$$V_{PCD}\left(t\right)=\hat{V_{PCD}}\;\cdot \;e^{\frac{\omega }{2Q}t}\left[\frac{1}{2Q}\sin\left(\omega t\right)+\cos\left(\omega t\right)\right]$$


(16)
$$\begin{array}{c} V_{PCD}\left(T_{2}\right)=-\hat{V_{c}}.e^{-\frac{\omega }{2Q}\frac{\pi }{\omega }}\left[\frac{1}{2Q}\sin\left(\omega \;\cdot \;\frac{\pi }{\omega }\right)+\cos\left(\omega \;\cdot \;\frac{\pi }{\omega }\right)\right]\\ =\hat{V_{c}}\;\cdot \;e^{-\frac{\pi }{2Q}}=\hat{V_{c}}\;\cdot \;\gamma \end{array}$$



t∈ [$T_{1}$, $T_{3}$]In this interval, as $V_{c}$ < $∥V_{c}∥$, the rectifier is disconnected and $I_{load}$ and the current flowing through the rectifier $I_{rect}$ can be expressed by Equation (17)



(17)
$$\{\begin{array}{ccc} I_{load} & & =-I_{c}\\ & I_{rect} & =I_{p}-I_{mod-PSSHI}=\alpha \dot{X}-C\dot{V_{c}}-I_{mod-PSSHI}=0\; \end{array}$$



t∈ [$T_{3}$, $T_{4}$]In this interval, the voltage $V_{PCD}$ reaches $\hat{V_{c}}$ and the rectifier is connected. $I_{rect}$ equals $I_{load}$+ $I_{c}$ where $I_{c}$ is the current flowing through the rectifier capacitor. $I_{rect}$ flows through the rectifier to the rectifier capacitor and then to the load resistor. The related expression of $I_{load}$ is shown in Equation (19)



(18)
$$I_{load}=I_{rect}-I_{c}=I_{p}-I_{mod-PSSHI}-I_{c}$$



t∈ [$T_{1}$, $T_{4}$]The rectifying capacitor is assumed as sufficiently large, so that the output voltage in this interval is regarded as a constant value and the net current through *C* is equal to zero. According to this assumption, the sum of the current flowing through the load resistor $I_{load}$ matches the sum of the current extracted from the *PCD*
$I_{p}$ as shown by Equation (19). The output voltage $\hat{V_{c}}$ can be derived by integrating the current $I_{load}$ from the time $T_{1}$ to $T_{4}$.

(19)$$I_{load}=I_{p}-I_{mod-PSSHI}$$(20)$$\int_{T_{1}}^{T_{4}}\frac{\hat{V_{c}}}{R}\;dt=\int_{T_{1}}^{T_{4}}\left(\alpha \dot{X}-C\dot{V_{c}}-I_{mod-PSSHI}\right)\;dt$$(21)$$\hat{V_{c}}=\frac{2R_{L}\alpha \omega }{\left(R_{L}C_{p}\omega \left(1-\gamma \right)+\pi \right)}\hat{X_{m}}$$which is the expression of the output voltage from the proposed interface $\left(V_{mod-PSSHI}\right)$ as seen by Equation (22)
(22)$$V_{mod-PSSHI}=\frac{2R_{L}\alpha \omega }{R_{L}C_{p}\omega \left(1-\gamma \right)+\pi }X_{m}$$

Figure 8a,b display the simulation result of the load current, the output voltage and the power of the proposed SP-PSSHI interface respectively. The simulations were carried out using LTspice SPICE simulator software. Table 3 shows the adopted simulation parameters. The maximum output voltage and power reached are 6.2 V and 3.9 mW respectively.

The inductance $L_{psshi}$ forms a resonant *LC* circuit with the harvester, where $C_{P}$ is the equivalent capacitance of the *PCD*. The purpose of the resonant *LC* circuit is to flip the harvester voltage in a very short time when compared to half of the harvester period by by closing switches (SW1) or (SW2) at the end of each half cycle and thus changing the polarity of the voltage through the capacitor as seen in Figure 6.

At time $T_{1}$ where $V_{load}$ is at its maximum value $V_{max}$, the modified P-SSHI current $I_{modP-SSHI}$ can be calculated using Equation (23), where $R_{series}$ is the new added series resistance to the PSSHI circuit.
(23)$$I_{mod-PSSHI}=\frac{V_{max}}{\sqrt{{\left(R_{psshi}+R_{series}\right)}^{2}+{\left(\omega \;\cdot \;L_{PSSHI}\right)}^{2}}}$$

The expression of the output power from the P-SSHI interface $P_{out-PSSHI}$ can be obtained by injecting the output voltage expression in the power formula as seen in Equation (24).
(24)$$P_{mod-PSSHI}=\frac{V_{out-PSSHI}^{2}}{R_{L}}=\frac{4R_{L}\alpha^{2}\omega^{2}}{{\left(R_{L}C_{p}\omega \left(1-\gamma \right)+\pi \right)}^{2}}X_{m}^{2}$$

In order to calculate the value of optimal load resistance $R_{mod-PSSHI,opt}$, we take the Equation (24) in partial differential equation to *R* and equals to zero as shown by Equation (25).
(25)$$\frac{\alpha P}{\alpha R}=\frac{4\alpha^{2}\;\cdot \;{\left[\pi +\left(1-\gamma \right)C_{p}R_{L}\omega \right]}^{2}-\left(4\alpha^{2}R_{L}\right)\;\cdot \;2\;\cdot \;\left(1-\gamma \right)C_{p}\omega \;\cdot \;\left[\pi +\left(1-\gamma \right)C_{p}R_{L}\omega \right]}{{\left[\pi +\left(1-\gamma \right)C_{p}R_{L}\omega \right]}^{4}}\omega^{2}X_{m}^{2}=0$$

The optimal resistor $R_{mod-PSSHI,opt}$ can be then obtained by Equation (26)
(26)$$R_{mod-PSSHI,opt}=\frac{\pi }{C_{p}\omega \left(1-\gamma \right)}$$

The maximum power output can be calculated by substituting the optimal resistance value $R_{mod-PSSHI,opt}$ into Equation (24), as seen in Equation (27)
(27)$$P_{max}{|}_{R=R_{mod-PSSHI,opt}}=\frac{\alpha^{2}\omega^{2}}{\pi \left(1-\gamma \right)C_{p}\omega }X_{m}^{2}$$

The power losses in P-SSHI interfaces $P_{losses-PSSHI}$ can be written as shown in Equation (28):(28)$$P_{losses-PSSHI}=P_{loss-trig}+P_{loss-sw}+P_{loss-l}+P_{loss-rect}$$
where,

$P_{loss-trig}$: power losses related to the triggering circuit$P_{loss-sw}$: power losses related to the integrated switches$P_{loss-l}$: power losses related to the integrated inductor L and the added series resistance$P_{loss-rect}$: power losses related to the rectifier block

The efficiency of the P-SSHI circuit $\eta_{psshi}$ can be expressed as follows in Equation (29), where $P_{in-PSSHI}$ can be expressed by Equation (30):(29)$$\eta_{psshi}=100\times \frac{P_{out-PSSHI}}{P_{in-PSSHI}}$$
(30)$$P_{in-PSSHI}=P_{out-PSSHI}+P_{losses-PSSHI}$$

With the activation time being significantly low, the power loss in the active switch branch is expected to be significantly low. Thus, the overall power loss in the circuit will be highly influenced by the P-SSHI inductive path and the selected rectifier for the P-SSHI unit $P_{loss-rect}$.

## 5. Experimental Validation

The aim is to design a wide-band piezoelectric energy harvesting system able to harvest energy from human footsteps at low frequencies, and efficiently manage and store the harvested energy to serve as an autonomous power supply for an inertial measurement unit (IMU) MAX21105 requiring [1.7–3.6 V] and a power-down current of 1.5 μA. The system consists of three main blocks, a developed shoe insole with integrated diaphragm piezoelectric harvesters, the proposed self-powered energy management P-SSHI interface and a small coin cell battery. A designed armband with integrated PEts was also used during the experiments to prove the adaptability of the proposed interface with different types and shapes of piezoelectric harvesters.

A piezoelectric ceramic sensor from Murata Manufacturing [38], was used as an energy harvester for the experiments. In the first stage, the characterization of the harvester and the investigation of its open circuit voltage in response to frequency variation from 1 Hz to 10 Hz was carried out. An electromagnetic shaker (VebRobotron Type 11077, Dresden, German) was used to generate the mechanical vibration, which was controlled through a function generator and an acceleration sensor. The PEH has the best performance at a frequency of 7 Hz and a related acceleration of 0.1 g as displayed in Figure 9. The used setup in this investigation is shown in Figure 10. A maximum open-circuit output voltage of 10.8 V was reached for a frequency equal to 7 Hz. For a frequency equal to 1 Hz, the PEH showed a minimum output voltage of 10.0 V. These results prove the suitability of the use of ceramic piezoelectric sensors for energy harvesting from footsteps which are generally ranging in the frequency band of 1–4 Hz.

The next experiments were carried out using the designed piezoelectric insole where the PCDs are integrated with different foot sections, such as forefoot, midfoot and heel. The PEts distribution was justified by an investigation of the state-of-the-art human pressure distribution during walking as seen in Figure 11. The insoles were designed using two different types of PCDs from Murata manufacturing, with different shapes and electrode diameters.

The electrode diameter of the integrated PEHs in the used insole for this experimental study is 29 mm. A person with a weight of 59 kg participated in the experiment to validate the system. The self-powered insole was placed on the ground and connected to the proposed circuit so that the subject could stand on it, and perform walking motion to extract energy from real human footsteps (moderate walking speed).

The output voltage and power data were displayed on the oscilloscope and recorded under different load resistances ranging from 10 kΩ to 1 MΩ and three different load capacitors of 10 μF, 100 μF and 1 mF. The specification of the hardware components is shown by Table 4. Lower the value of the load capacitor, the higher the maximum output DC voltage and output power achieved at the load. The maximum output power of 3.6 mW and related output voltage of 6.0 V was reached at a loading capacitor and resistance of 10 μF and 10 kΩ respectively (see Figure 12).

The corresponding input voltage is 7.2 Volts peak to peak (Vpp). Under the same load capacitor of 10 μF and a load resistor of 1 MΩ, the minimum output power of 0.125 mW was reached. Figure 13 illustrates the used experimental setup.

The supplied current of the SP-PSSHI circuit to the load as well as the interface efficiency was also recorded under the same conditions. A maximum supply current and efficiency of 2.1 μA and 83.33% were reached respectively under the optimum loading conditions of 10 μF capacitor and 10 kΩ resistor (see Figure 14). The minimum interface efficiency of 69.6% was reached when the loading resistor is 47 kΩ. The related output power and voltage are 0.87 mW and 6.4 V respectively.

The second experiments were conducted at the selected optimum loading resistor of 10 kΩ and aimed to test the proposed wearable energy harvesting system in order to perform energy storage with different storage capacitors of 10 μF, 100 μF, 220 μF and 1 mF in a first stage and to charge a coin cell battery in a second stage as shown in Figure 15. The charging time and stored voltage data were recorded. We figured out that the higher the value of the load capacitor, the longer it takes time to ramp up the DC output voltage. The proposed system was validated to charge a 10 μF capacitor from 0.8 V to 4.8 V in 2.71 s, a 100 μF capacitor from 0.8 V to 3.6 V in 12 s and a 1 mF capacitor from 0.8 V to 3.2 V in 27.64 s.

The proposed interface was also validated to charge a (65 mAh, 3.1 V) Lithium Maganese dioxide (ML 2032) coin cell battery commonly used to supply wearable biomedical sensors from 1.4 V to 1.76 V in 40 min as displayed in Figure 16.

The charge and discharge of the device between two close energy levels were carried out. When the storage device’s voltage reached a reference voltage $V_{0}$ during the piezoelectric harvesting current charging process, time calculation started. After a certain time, interval $t_{1}$, the voltage increases by Δ*V*. At the end of the interval *t*_1_, the cell charging stopped and a known discharge current value $I_{dis}$ was used to discharge the cell for a time interval *t*_2_ to a voltage $V_{0}-\Delta$*V*. At the end of time interval $t_{2}$, the cell discharging stopped and start to recharge to a voltage $V_{0}$ starts. $t_{3}$ is denoted as the time interval to recharge the cell to voltage $V_{0}$ again. To test the charge/discharge efficiency of the energy storage process, a charger is connected to the storage device until fully charged. Then, a load is connected to the energy storage device to fully discharge it. The following Equation (31) is used to calculate the energy storage device’s efficiency.
(31)$$\eta =\frac{DischargeEnergy}{ChargeEnergy}$$

The charge current can be calculated using the following Equation (32).
(32)$$I_{charge}=\frac{\left(V_{max}-V_{OC}\right)}{BatteryResistance}$$
where $V_{max}$ is the maximum safe voltage of the battery and $V_{OC}$ is the open circuit voltage of the battery. The charging time can be given by Equation (33):(33)$$T_{charge}=\frac{Capacity(\mathrm{Ah})}{ChargeRate(A)}$$

The discharge rate of the battery can be given by Equation (34):(34)$$Discharge\;Rate=\frac{Battery\;Capacity(\mathrm{Ah})}{T_{charge}/T_{discharge}\left(h\right)}$$

The major factor influencing the charge or discharge efficiency of a storage device in a piezoelectric energy harvesting system (low current application) is the leakage resistance. Therefore, storage energy devices with high leakage resistance are preferred. Storage capacitors offer higher leakage resistance in comparison with Lithium batteries and NiMH batteries. To conclude, Table 5 compares the previously reviewed piezoelectric energy harvesting circuits in the literature with the proposed work. The explored up to date state of the art evaluates the developed interfaces mainly in terms of harvested output power and the implemented storage device with the related stored voltage and charging time. The frequency range and the loading conditions under which the interfaces were evaluated were also taken into consideration.

## 6. Conclusions

A full wearable piezoelectric energy harvesting system was proposed, including the harvester stage, the energy management interface, and the storage device. In order to validate the performance and output power of the proposed solution, a prototyping piezoelectric insole was designed with six parallel PEHs. A self-powered GCC-PSSHI energy management circuit was designed to harvest energy from piezoelectric converters at low frequencies and irregular shock-like footstep input excitations. Using the designed insole, a 10 kΩ resistive load, and a 10 μF storage capacitor, the maximum efficiency was 83.02% and the output power was 3.6 mW. The proposed solution was validated to charge a 1 mF capacitor to 3.2 V in 27.64 s and a 10 μF capacitor to 6 V in 3.94 s using the designed insole. The proposed solution was also validated to charge a (65 mAh, 3.1 V) ML 2032 coin cell Lithium battery to demonstrate the ability to implement the proposed wearable piezoelectric energy harvesting system to provide an autonomous power supply for body-attached sensors. In addition to being self-powered, the proposed approach is built of a small number of components that consume low energy, has a cold start-up ability, can harvest energy from low frequency and irregular footsteps input excitations, which typically result from walking motion, and maintains a constant efficiency at low input voltages.

## Figures and Tables

**Figure 1 sensors-23-01830-f001:**
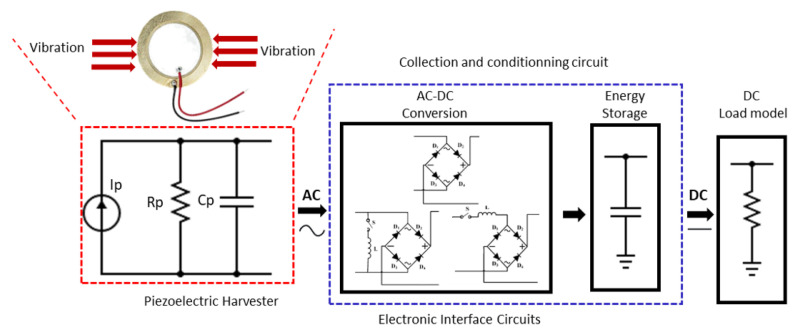
Block diagram of PEH system.

**Figure 2 sensors-23-01830-f002:**
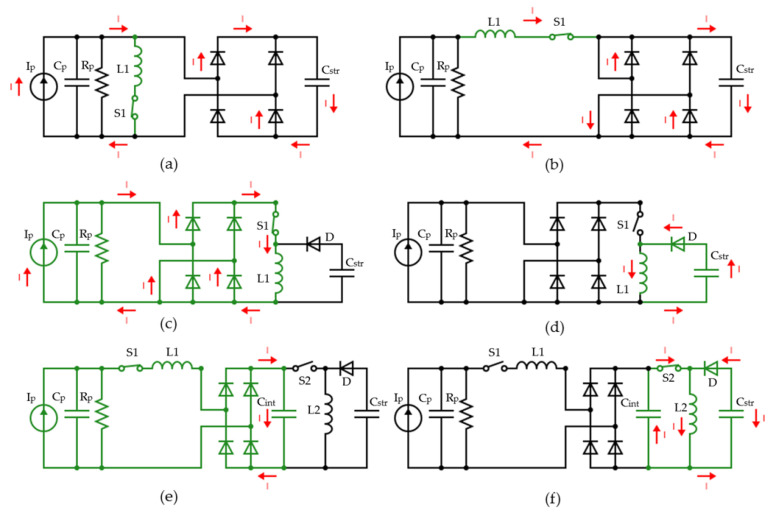
(**a**) Schematic of P-SSHI energy extraction interface, (**b**) schematic of S-SSHI energy extraction interface, (**c**) Schematic of SECE energy extraction interface when the switch S is closed, (**d**) schematic of SECE energy extraction interface when the switch S is open, (**e**) Schematic of D-SSHI energy extraction interface when the switch S1 is closed and S2 is opened, (**f**) schematic of D-SSHI energy extraction interface when the switch S1 is opened and S2 is closed [29].

**Figure 3 sensors-23-01830-f003:**
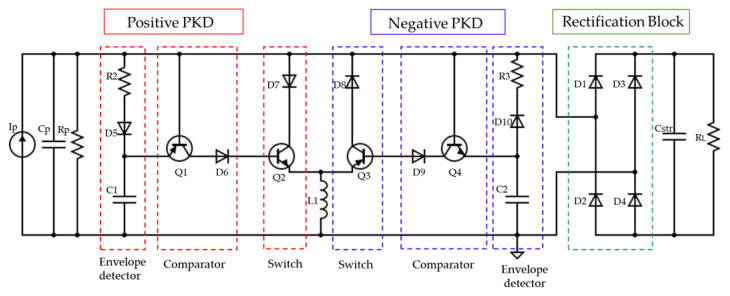
Schematic of the simulation circuit of the state-of-the-art SP-PSSHI referring to [30].

**Figure 4 sensors-23-01830-f004:**
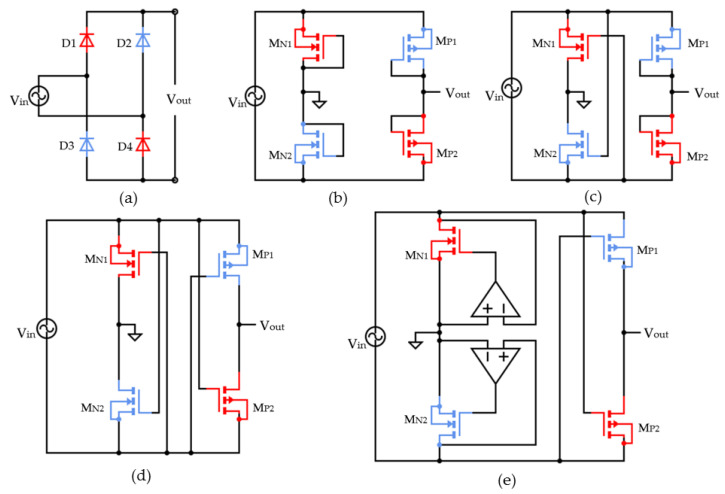
AC-DC rectifiers for SSHI interfaces: (**a**) Diode Bridge, (**b**) MOSFET Bridge (2N & 2P), (**c**) GCCR (2N & 2P), (**d**) NVC (2N & 2P), (**e**) Active Rectifier with Cross-Coupled PMOS Switches (2N & 2P) [28].

**Figure 5 sensors-23-01830-f005:**
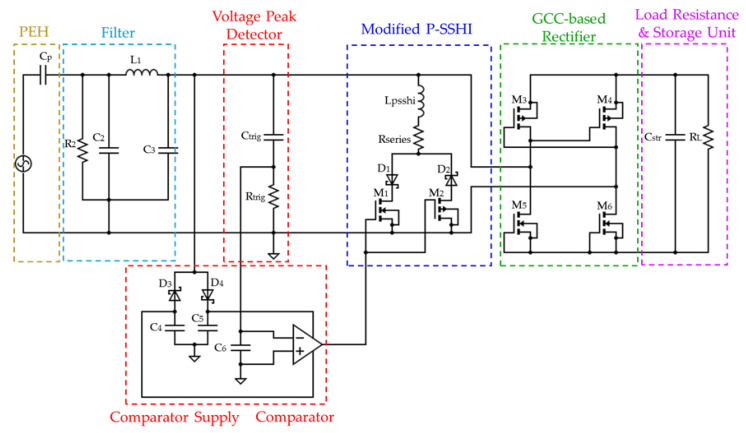
Proposed SP-PSSHI energy harvesting circuit model.

**Figure 6 sensors-23-01830-f006:**
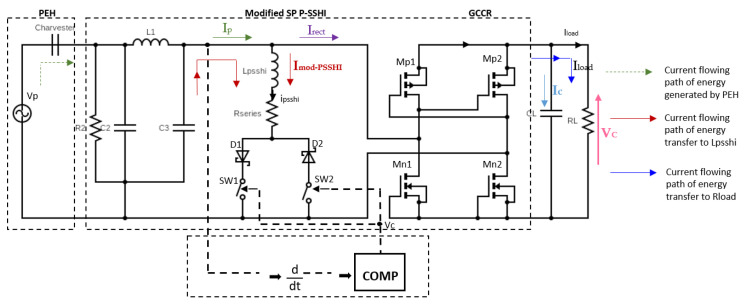
Schematic design concept of the proposed SP-PSSHI energy harvesting approach.

**Figure 7 sensors-23-01830-f007:**
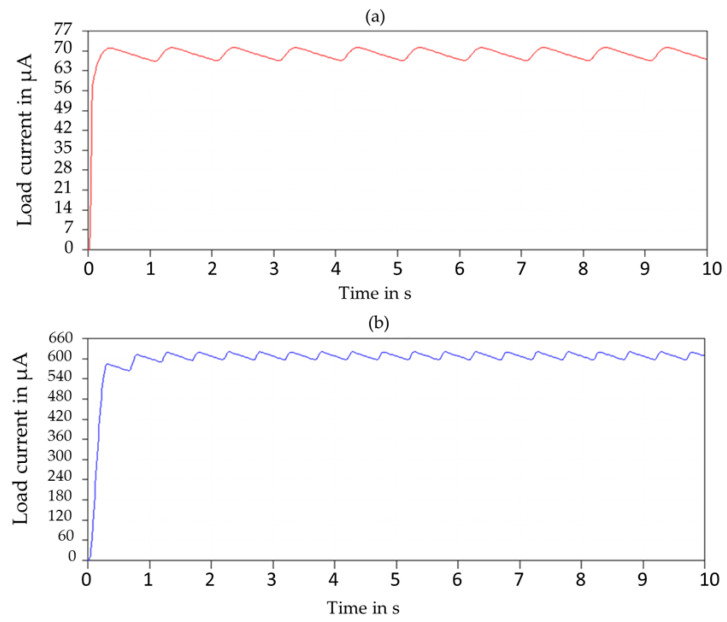
Simulated output current flowing through the load (**a**) before the added series resistance (**b**) after the added series resistance.

**Figure 8 sensors-23-01830-f008:**
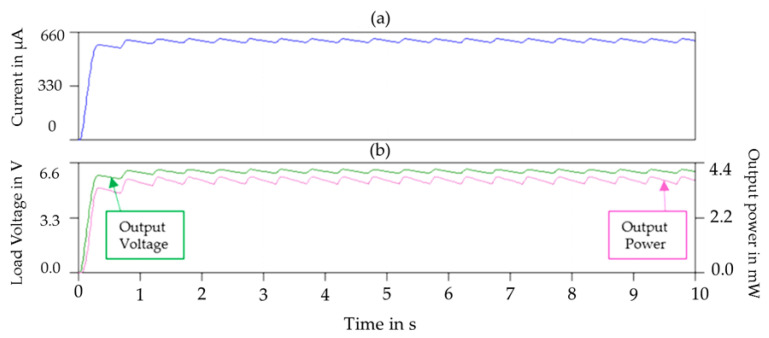
(**a**) Simulated Load current at R = 10 kΩ, (**b**) Simulated output voltage and output power of the proposed SP-PSSHI circuit.

**Figure 9 sensors-23-01830-f009:**
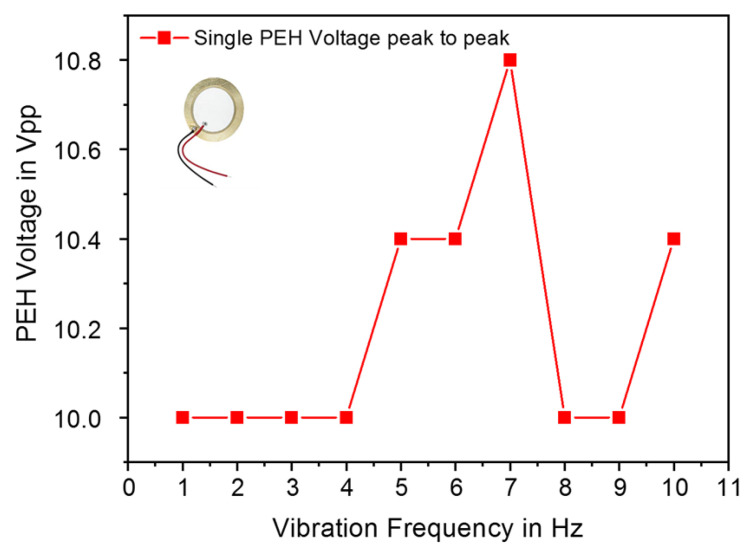
Piezoelectric Harvester open circuit voltage versus frequency variation.

**Figure 10 sensors-23-01830-f010:**
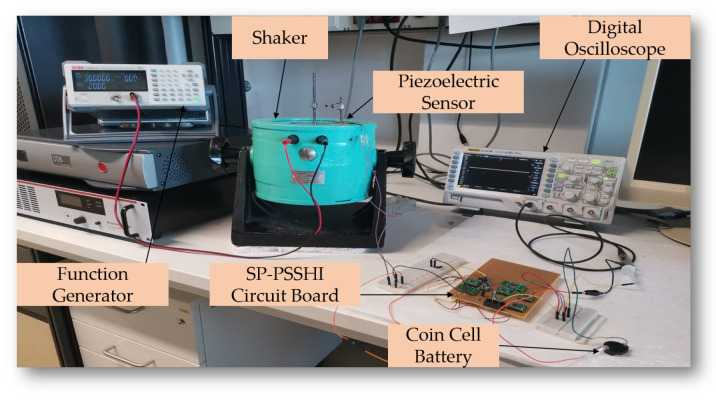
Experimental setup for piezoelectric harvester characterization and battery charging process.

**Figure 11 sensors-23-01830-f011:**
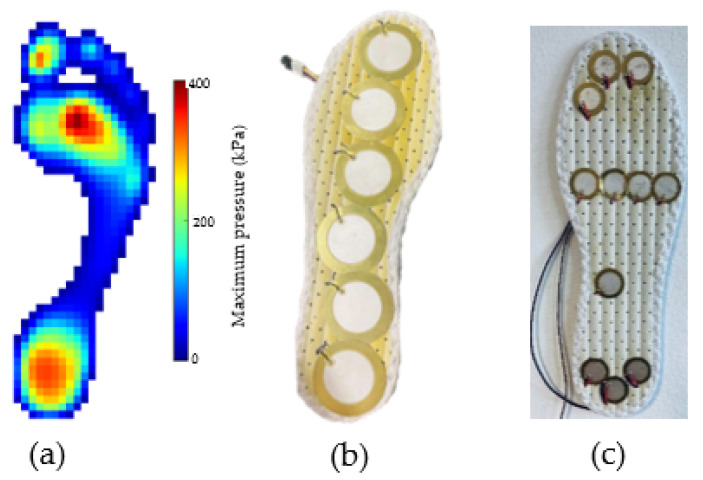
(**a**) Footsteps pressure distribution [47], (**b**,**c**) PEts distribution in the designed prototyping piezoelectric insoles.

**Figure 12 sensors-23-01830-f012:**
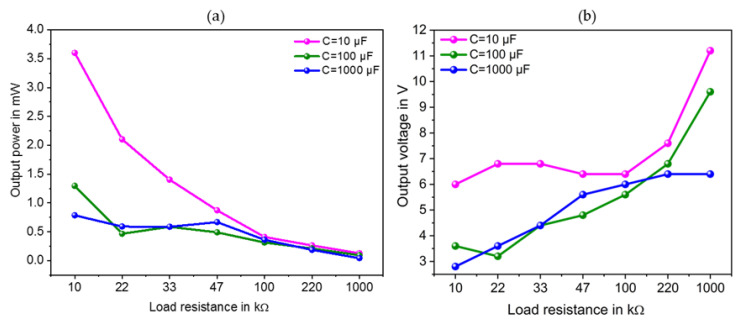
Output power and output voltage versus load resistance through different load capacitors (**a**,**b**) for the proposed SP-PSSHI interface.

**Figure 13 sensors-23-01830-f013:**
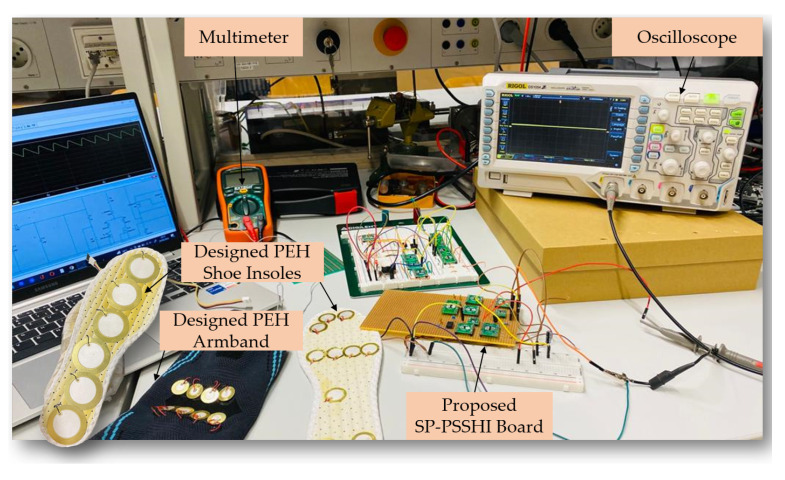
Experimental setup for laboratory characterization of the proposed energy management interface and storage process.

**Figure 14 sensors-23-01830-f014:**
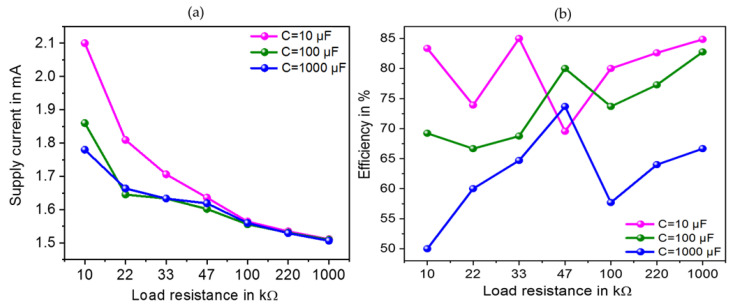
Current supplied by piezoelectric device and the output efficiency versus load resistance through different load capacitors (**a**,**b**).

**Figure 15 sensors-23-01830-f015:**
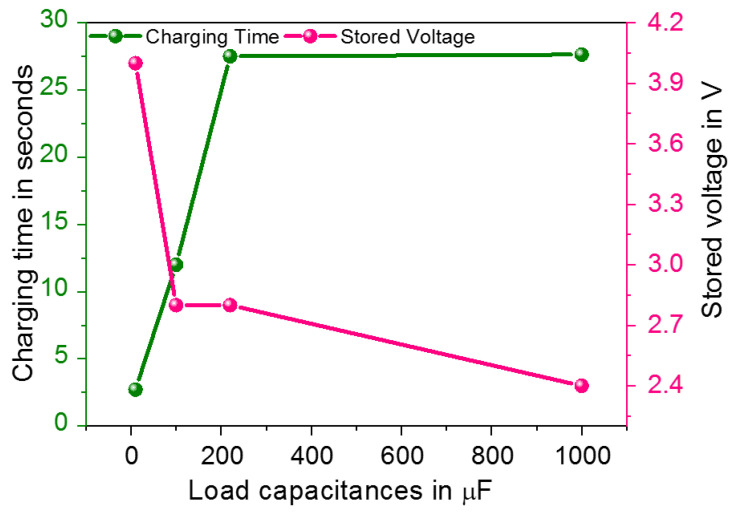
DC Voltage stored and related charging time versus load capacitance.

**Figure 16 sensors-23-01830-f016:**
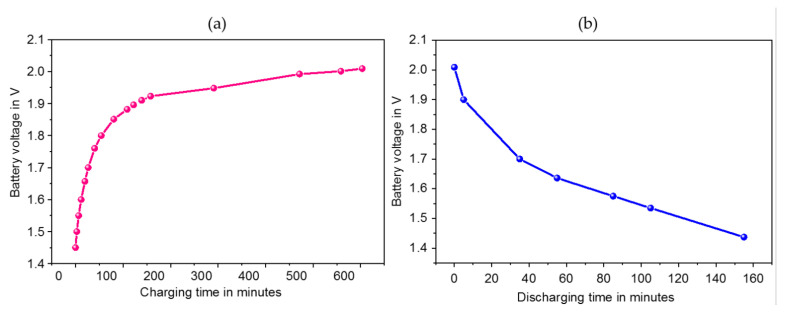
(**a**) Charging time of the Lithium coin cell battery versus stored voltage (**b**) Discharging time versus the stored battery voltage.

**Table 1 sensors-23-01830-t001:** Influence of the rectifier type on the performance of State of the Art P-SSHI for resistive loads [28].

Interface	Input Voltage (V)	Input Power (mW)	Output Power (mW)	Efficiency (%)
MOSFET Bridge	2	0.315	0.158	50.08
GCCR		0.479	0.35	73.04
NVC		0.664	0.662	99.77

**Table 2 sensors-23-01830-t002:** Influence of the rectifier type on the performance of State of the Art P-SSHI for RC loads [28].

Interface	Input Voltage (V)	Input Power (mW)	Output Power (mW)	Efficiency (%)
Diode Bridge	0.3–0.5	0.75	0.15	25
MOSFET Bridge	2	0.623	0.306	49.07
GCCR		0.949	0.692	72.99
NVC		34.6	0.784	0.024

**Table 3 sensors-23-01830-t003:** Simulation parameters values.

Parameter	Symbol	Value
Frequency	*f*	1 Hz
PEt capacitance	$C_{p}$	0.15 μF
Inductance	$L_{psshi}$	10 mH
Added series resistance	$R_{series}$	10 KΩ
Load resistance	$R_{L}$	10 KΩ
Load capacitance	$C_{L}$	1 mF

**Table 4 sensors-23-01830-t004:** Specifications of hardware components.

Main Component	Reference	Characteristics
Diodes	BAT54	$V_{f}$ = 0.4 V, $I_{f}$ = 10 mA
MOSFETs (N channel)	IRLML6246TRPBF (Infineon Technologies)	$R_{DS(ON)}$ = 46 mΩ Vg(th) = 0.5 V
MOSFETs (P channel)	AO3435	$R_{DS(ON)}$ = 70 mΩ $V_{g(th)}$ = −0.5 V
Comparator	TLV3691 (Texas Instrument)	$V_{S}$ = 0.9–6.5 V $I_{Q}$ = 75 nA

**Table 5 sensors-23-01830-t005:** Comparaison of the proposed SP PSSHI Circuit with State of the art solutions.

Ref.	[48]	[49]	[49]	[50]	[34]	This Work
Architecture	SEH	VM (0.65 V)	SEH (Vf 0.55 V)	SECE	NVC-PSSHI	SP GCC-PSSHI
Frequency	0.6 Hz	8.9 Hz	8.9 Hz	1 Hz	100 Hz	1 Hz
Switch Integration	No	No	No	YES	YES	YES
Piezoelectric Harvester	Flexible BLFPT/PI	MFC M281-P2 (d31), Smart material	macro-fiber composite (MFC)	commercial PPA1011 PEH (single beam)	PPA-1011 Mide Tech	AB4113B- LW100-R
Storage Device	C storgae 1 μF C storage 4.3 mF	C storage 10 μF	C storage 10 μF	C storage 47 μF	C storage 1 mF 40 mAh Lipo battery	C storage 10 μF C storage 1 mF
Stored Voltage	1.5 V 1.6 V	2.5 V	4.156 V	3.8 V	3.6 V 0.24 V	6.0 V 3.6 V
Charging Time	150 s 300 s	NA	NA	NA	400 s 14 Days	3.94 s 27.64 s
Load Resistance	NA	100 kΩ	100 kΩ	NA	No load	10 KΩ (83.3 %)
Output Power	NA	62.3 μW	172.2 μW	10.2 μW	300 μW for 3–7 Vpp	3.6 mW for 7.2 Vpp
Technology	Discrete components	Discrete components	CMOS technology	CMOS technology	Discrete components	Discrete components

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
