# Peer review of "Self-Powered Synchronized Switching Interface Circuit for Piezoelectric Footstep Energy Harvesting"

_sensors, 2023, doi:10.3390/s23041830_

Round 1

Reviewer 1 Report

This paper systematically introduces the necessity and potential problems of SSHI in piezoelectric energy harvesting, and proposes a targeted improvement plan. The modified scheme proves the effectiveness of this work through the combination of simulation and experiment. However, this is a research work, but at this stage the overall structure of the article is more like a comprehensive graduation thesis. Therefore, I suggest modifying the layout of the pictures in this article and making the text more compact. I recommend accepting this paper after a major revision.

1.      There are overall 22 figures in this paper. I suggest that by optimizing and merging your results, try to control the number of large graphs, and try to explain as much as possible around one problem in one large figure.

2.      The language of the article needs to be refined.

3.      “PZT” usually only presents the material with the element components: Pb1ZrxTi1-xO3. Please refer to the following papers: https://doi.org/10.1021/acsami.0c16973, https://doi.org/10.1016/j.pmatsci.2018.12.005 . Thus, I suggest not using PZT to abbreviate the “piezoelectric sensors” in the paper.

4.      In line 264, there is no relation between flexible piezoelectric devices and biocompatibility. The material itself determines the toxicity. Please refer to the following paper: https://doi.org/10.1002/adma.201802084 . I suggest citing a series of papers to facilitate the description on piezoelectric energy harvesters, such as:  https://doi.org/10.1016/j.nanoen.2020.105567.

5.      In section 5, the aim of the experiment is to harvest energy from footstep. The author tested the frequency from 1 Hz to 10 Hz. Please include the specific parameters of the vibration, for example, the amplitude, the acceleration rate, etc. And the fixture methods may also affect the results.

6.      In figure 16a, how do the author obtain the footstep pressure distribution?

7.      In figure 17, the resistance range seems not enough to reveal the best impedance matching value. There is usually a peak value of output power at a certain resistance value.

Author Response

Dear reviewer,

please find all corrections in the attached pdf file. 

Regards.

Reviewer 2 Report

See attached.

Author Response

(The authors gave the same response as above.)

Reviewer 3 Report

The paper describes a methodology to harvest energy from irregular chock-like footstep input excitations. The author proposed a Self-Powered (SP P-SSHI) energy management circuit capable to harvest energy from piezoelectric converters at low frequencies. The authors developed and analytical model to optimize and maximize the harvested power as well as increase the output voltage. They created an experimental setup to evaluate the performance of their suggested method. They compared this method to the previous prior arts and concluded that the newly developed method results in higher harvested power

The paper is well written and the literature review is well developed, there are some minor grammatical errors that need to be fixed like

Line 458, there is a unit issue, it should be ampere not Farad 

Line 490, The lower

Line 503, Carried out

In paragraph 510 to 525, avoid using we, try to use past tense instead

530 The leakage resistance

Author Response

(The authors gave the same response as above.)

Round 2

Reviewer 1 Report

I think it is good to publish now. 

Reviewer 2 Report

Accept as it is